# Prediction of Stress and Deformation Caused by Magnetic Attraction Force in Modulation Elements in a Magnetically Geared Machine Using Subdomain Modeling

**Manh-Dung Nguyen** [1], **Su-Min Kim** [1], **Jeong-In Lee** [2], **Hyo-Seob Shin** [1], **Young-Keun Lee** [1], **Hoon-Ki Lee** [1], **Kyung-Hun Shin** [3], **Yong-Joo Kim** [4], **Anh-Tuan Phung** [5] and **Jang-Young Choi** [1,*]

1   Department of Electrical Engineering, Chungnam National University, Daejeon 34134, Republic of Korea; nguyenmanhdung@o.cnu.ac.kr (M.-D.N.); su.min@o.cnu.ac.kr (S.-M.K.); shs1027@cnu.ac.kr (H.-S.S.); 201502206@o.cnu.ac.kr (Y.-K.L.); lhk1109@cnu.ac.kr (H.-K.L.)
2   Electric Power System Design Team, Hyundai Transys Inc., Hwaseong 18463, Republic of Korea; jilee@hyundai-transys.com
3   Department of Power System Engineering, Chonnam National University, Yeosu 59626, Republic of Korea; kshin@chonnam.ac.kr
4   Department of Bio-Systems Machinery Engineering, Chungnam National University, Daejeon 34134, Republic of Korea; babina@cnu.ac.kr
5   Department of Electrical Engineering, Hanoi University of Science and Technology, Hanoi 100000, Vietnam; tuan.phunganh1@hust.edu.vn
*   Correspondence: choi_jy@cnu.ac.kr

**Abstract:** This study presents an approach for calculating the stress and deformation increase in the modulation of magnetically geared machines using a mathematical method. An analytical method is employed to obtain the magnetic force density acting on the modulation components. Afterward, the proposed mathematical method predicts the mechanical characteristics. The 9 slots/32 poles/19 modulations model was evaluated via a comparison with the finite element method simulation.

**Keywords:** magnetically geared machine; partial differential equations; subdomain modeling; analytical solution; FEM





## 1. Introduction

Currently, there are two approaches for addressing high-torque, low-speed applications. The first approach is a combination system of a gear and small-pole machine, which takes up more space. The second one entails using a large-pole-number machine that results in a complicated winding organization and increases the manufacturing cost. Both approaches have drawbacks. Hence, magnetically geared machines (MGMs) have become an increasingly attractive option for integrating permanent magnet synchronous machines (PMSMs) with magnetic gears into one structure. A comprehensive overview of the evolution and various approaches to MGMs has been presented [1–3]. Four different winding organizations for MGMs were investigated in [4,5], in which the authors proposed a dual-structure machine that improved the torque and reduced the cogging torque. To obtain the air gap flux density, an analytical method was employed [6]. Performance analyses of double-rotor or double-stator MGMs were also carried out in [7–12], where the authors presented a novel "pseudo" machine whose architecture is similar to a magnetic gear, in which the outer rotor has attached a winding having the same pole pair with the inner rotor. Using a spoke-type magnet [13–15] was an alternative to surface-type magnets [16] in the Vernier structure. Through changing the shape of the inner permanent magnet and the magnetization of the outer permanent magnet, a new structure has high torque density and high mechanical properties [17–19]. Combinations of spoke-type and Halbach-type

rotors, as well as auxiliary teeth in the stator, were introduced [20,21]. A novel approach involving double-fed current is illustrated in [22–25], in which the authors introduced a novel bidirectional flux modulation concept, and [26] merged modulation and a stator into a single structure based on bidirectional concepts. To deepen comprehension and amplify output torque, investigations were carried out regarding gear ratios [27], parameter analysis [28], and optimization [29]. Notably, the eccentric harmonic magnetic gear was introduced in [30], proving particularly advantageous for applications demanding higher transmission ratios.

The aforementioned studies mainly focused on the electromagnetic aspect and superficially dealt with the mechanical analyses. In [31], the authors utilized FEM, which required several hours to obtain results. Ref. [32] experimentally verified a laminated structure in modulation after simulating a relevantly simplified model.

The FEM is widely recognized as a reliable solution for electrical machine analysis but has a significant running time when calculating problems. The subdomain analytical method, on the other hand, offers faster computational time and results that agree well with FEM simulations, making it a promising solution for MGM design and optimization. A comprehensive review can be found in the literature [33]. Two analytical methods are mainly used. The first one is based on the conformal mapping transformation method and a 2D relative permanence function [34]. The second method, which can be named subdomain modeling, is based on directly solving Maxwell's equations and the boundary and interface conditions. Researchers have applied the subdomain method to various scenarios, such as radial-magnetization surface-mounted PMSM, Halbach array PMSM, coaxial magnetic gear, axial-field magnetic gear, and eccentricity effects in magnetic gear, as demonstrated in [35–39]. To consider iron parts with nonlinear cores, an enhanced 2D subdomain method in polar coordinates was explored [40,41].

The purpose of this paper is to present a 2D analytical model of a magnetic gear machine (MGM) based on the subdomain method. To the best of our knowledge, the analytical prediction of stress and deformation for a solid-structured modulation has not been explored previously. To address this gap, the authors utilized subdomain modeling to obtain the magnetic force density and proposed a purely mathematical approach for analyzing stress and deformation.

## 2. Magnetic Attraction Force

### 2.1. Governing Partial Differential Equations (PDEs)

Figure 1 presents an MGM model and its simplified reverent transfer model employed for the analytical method. Tables 1 and 2 list the parameters of the machine and materials. Before solving the analytical solutions, the following assumptions are made:

- The end effects are ignored;
- The problem is two-dimensional in Cartesian coordinates;
- Magnetic vector potential $A$, current density $J$, magnetization vector $M$, and magnetic flux density vector B, have the following non-zero components, respectively: $A = [0, 0, A_z]$; $J = [0, 0, J_z]$; $M = [M_r, M_\theta, 0]$; $B = [B_r, B_\theta, 0]$;
- The core materials have infinite permeability;
- The shaft is a non-magnetic material.

**Table 1.** The mechanical coefficients.

| Parameters | Symbol | Unit | Value |
|---|---|---|---|
| Young's modulus | $E$ | $10^5$ MPa | 2 |
| Shear ratio | $\kappa$ | - | 0.85 |
| Bulk modulus | $G$ | MPa | 76,923 |

**Table 2.** Specification parameters.

| Parameters | Symbol | Unit | Value |
|---|---|---|---|
| Outer magnet radius | $R_7$ | mm | 93.5 |
| Inner magnet radius | $R_6$ | mm | 90.0 |
| Outer modulation radius | $R_5$ | mm | 89.0 |
| Inner modulation radius | $R_4$ | mm | 81.0 |
| Outer stator radius | $R_3$ | mm | 80.0 |
| Opening slot radius | $R_2$ | mm | 78.5 |
| Slot radius | $R_1$ | mm | 37.0 |
| Stack length | $L_{stk}$ | mm | 45.0 |
| Magnet pitch ratio | $\alpha$ | - | 0.9 |
| Modulation pitch | $\gamma$ | $2\pi/P_m$ rad | 0.5 |
| Opening slot pitch | $\beta$ | $2\pi/Q$ rad | 0.125 |
| Slot pitch | $\delta$ | $2\pi/Q$ rad | 0.8 |
| Pole winding turn | - | - | 56 |
| Modulation number | $P_m$ | - | 19 |
| Slot number | $Q$ | - | 9 |
| Magnet pole pair | $Z_p$ | - | 16 |
| Vacuum permeability | $\mu_0$ | $kg.m.s^{-2}.A^{-2}$ | $4\pi\,10^{-7}$ |
| Residual flux density of magnet | $B_0$ | T | 1.2 |
| Magnetic magnetization of magnet | $M_0$ | A/m | $B_0/\mu_0$ |

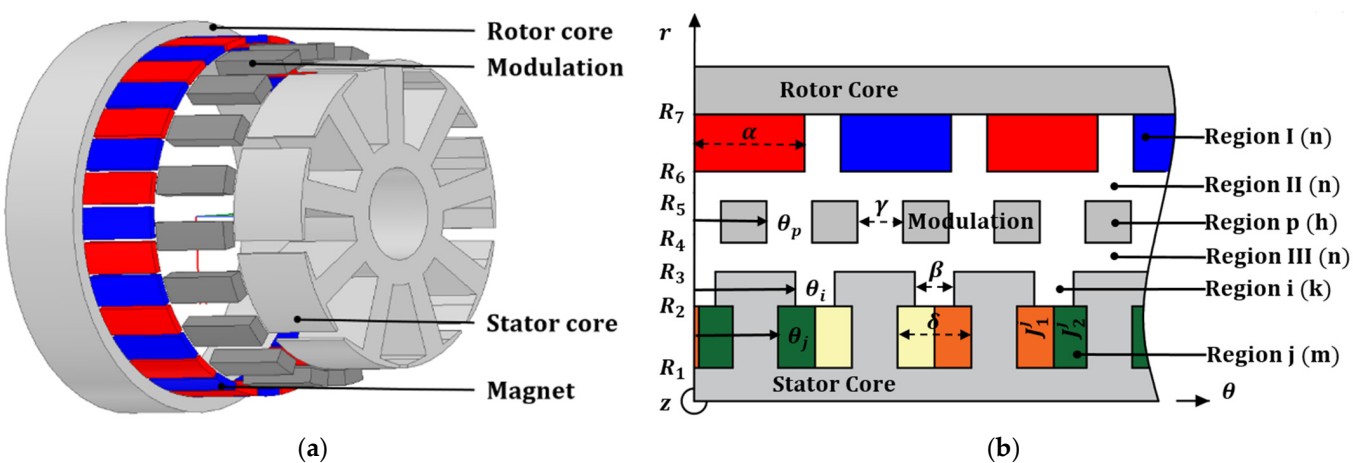

**Figure 1.** (**a**) Configuration and (**b**) simplified model of the MGGs.

Through assuming 2D analysis in cylindrical coordinates, the governing partial differential equations (PDEs) for the six regions shown in Figure 1 are expressed as follows:

$$\frac{\partial^2 A_z^I}{\partial r^2} + \frac{\partial A_z^I}{r\partial r} + \frac{\partial^2 A_z^I}{r^2\partial\theta^2} = -\frac{\mu_0}{r}\left(M_\theta^I - \frac{\partial M_r^I}{\partial\theta}\right) \tag{1}$$

$$\frac{\partial^2 A_z^{II}}{\partial r^2} + \frac{\partial A_z^{II}}{r\partial r} + \frac{\partial^2 A_z^{II}}{r^2\partial\theta^2} = 0 \tag{2}$$

$$\frac{\partial^2 A_z^p}{\partial r^2} + \frac{\partial A_z^p}{r\partial r} + \frac{\partial^2 A_z^p}{r^2\partial\theta^2} = 0 \tag{3}$$

$$\frac{\partial^2 A_z^{III}}{\partial r^2} + \frac{\partial A_z^{III}}{r\partial r} + \frac{\partial^2 A_z^{III}}{r^2\partial\theta^2} = 0 \tag{4}$$

$$\frac{\partial^2 A_z^i}{\partial r^2} + \frac{\partial A_z^i}{r\partial r} + \frac{\partial^2 A_z^i}{r^2\partial\theta^2} = 0 \tag{5}$$

$$\frac{\partial^2 A_z^j}{\partial r^2} + \frac{\partial A_z^j}{r\partial r} + \frac{\partial^2 A_z^j}{r^2\partial\theta^2} = -\mu_0 J_z^j \tag{6}$$

It can be seen that the subdomains of modulation, air gap, and opening slot are represented using Laplace's equation, whereas the subdomains of magnet and slot are described using Poisson's equation. To apply the Fourier series expansion method to the vector potential, it is necessary to express the right-hand side of Poisson's equation in Fourier form.

Figure 2 depicts the parallel magnetization at an arbitrary position of the rotor, denoted as $\theta_0$. In the 2D polar coordinate system, the magnetic magnetization can be expressed as follows:

$$\boldsymbol{M_z^I} = \sum_{n=1,2}^{\infty} M_{rn}^I \sin(n(\theta-\theta_0))\boldsymbol{i_r^I} + \sum_{n=1,2}^{\infty} M_{\theta n}^I \cos(n(\theta-\theta_0))\boldsymbol{i_\theta^I}$$

where

$$M_{rn}^I = -\frac{2M_0}{\pi}\sum_{p=1,2}^{2Z_p}\frac{(-1)^i}{n+1}\sin\frac{(n+1)\pi\alpha}{2Z_p}\sin\frac{n\pi(i-0.5)}{Z_p} - \begin{bmatrix} \frac{2M_0}{\pi}\sum_{p=1,2}^{2Z_p}\frac{(-1)^i}{n-1}\sin\frac{(n-1)\pi\alpha}{2Z_p}\sin\frac{n\pi(i-0.5)}{Z_p} \leftrightarrow n \neq 1 \\[2ex] \frac{-2M_0}{\pi}\sum_{p=1,2}^{2Z_p}(-1)^i\frac{\pi\alpha}{2Z_p}\sin\frac{n\pi(i-0.5)}{Z_p} \leftrightarrow n = 1 \end{bmatrix} \tag{7}$$

$$M_{\theta n}^I = \frac{2M_0}{\pi}\sum_{p=1,2}^{2Z_p}\frac{(-1)^i}{n+1}\sin\frac{(n+1)\pi\alpha}{2Z_p}\sin\frac{n\pi(i-0.5)}{Z_p} - \begin{bmatrix} \frac{2M_0}{\pi}\sum_{p=1,2}^{2Z_p}\frac{(-1)^i}{n-1}\sin\frac{(n-1)\pi\alpha}{2Z_p}\sin\frac{n\pi(i-0.5)}{Z_p} \leftrightarrow n \neq 1 \\[2ex] \frac{-2M_0}{\pi}\sum_{p=1,2}^{2Z_p}(-1)^i\frac{\pi\alpha}{2Z_p}\sin\frac{n\pi(i-0.5)}{Z_p} \leftrightarrow n = 1 \end{bmatrix}$$

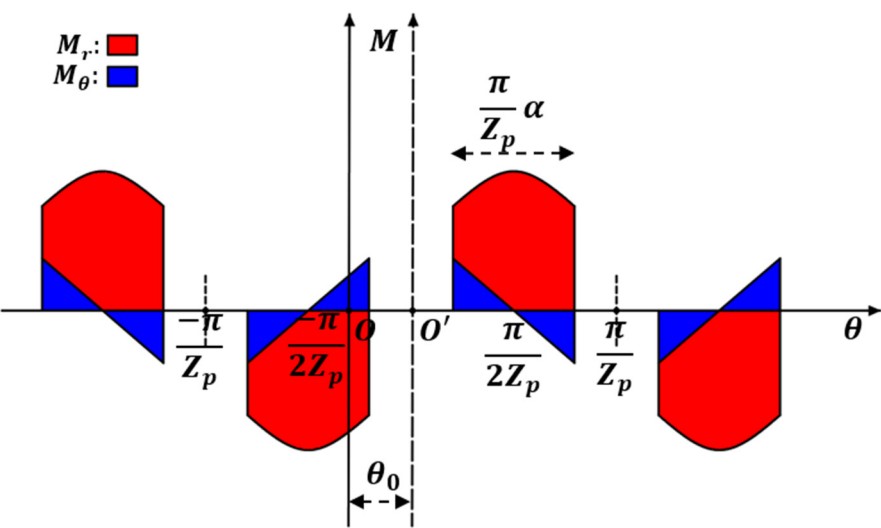

**Figure 2.** The parallel magnetic magnetization illustration.

Regarding coil magnetization, when using a double-layer winding, there are two methods of winding organization: overlapping winding for distributed winding and non-overlapping winding for concentrated winding. Figure 3 provides an illustration of how the coil is accommodated in the slot for each of these methods.

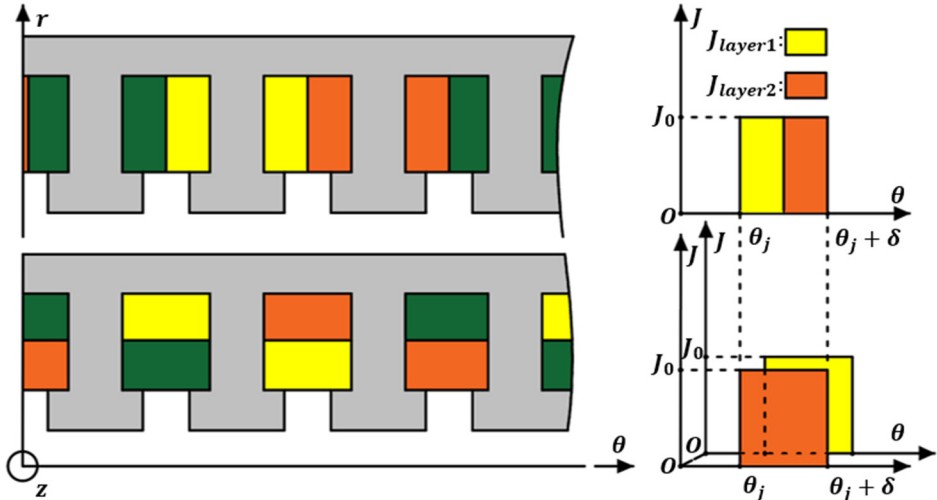

**Figure 3.** Winding layouts: non-overlapping (top) and overlapping (bottom) windings.

In either case, the current density of the *j*-th order slot can be represented in Fourier form.

$$J_z^j = \left( J_0^j + \sum_{m=1,2}^{\infty} J_m^j \cos\left(\frac{m\pi}{\delta}(\theta - \theta_j)\right) \right) i_z$$

where

$$J_0^j = \frac{J_0^{j-layer1} + J_0^{j-layer2}}{2} \tag{8}$$

$$J_m^j = \left[ \begin{array}{l} \frac{2}{m\pi}\left(J_0^{j-layer1} - J_0^{j-layer2}\right)\sin\left(\frac{m\pi}{2}\right) \leftrightarrow non-overlapping\ winding \\ \qquad\qquad 0 \leftrightarrow overlapping\ winding \end{array} \right.$$

### 2.2. Boundary Conditions Principle

In Figure 4 without magnet excitation, the boundary condition between two adjacent materials is applied based on three principles: $H_{\theta 1} = H_{\theta 2}$, $B_{r1} = B_{r2}$, and $A_{z1} = A_{z2}$. For mathematical convenience, it is preferable to choose the first and third principles, and rewrite them in terms of vector potential as follows:

$$\frac{\partial A_{z1}}{\mu_1 \partial r} = \frac{\partial A_{z2}}{\mu_2 \partial r} \tag{9}$$

$$A_{z1} = A_{z2} \tag{10}$$

where $\mu_1$ and $\mu_2$ are the relative permeability of the first and second materials, respectively.

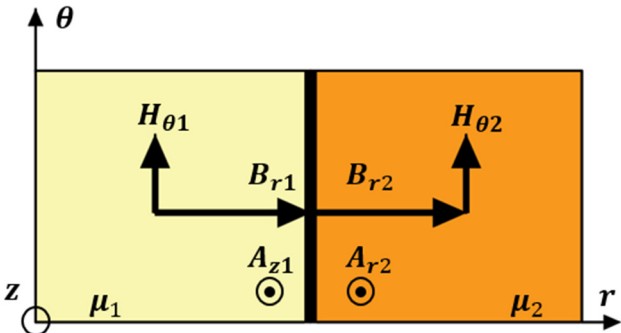

**Figure 4.** Boundary condition principle.

If the first material is the magnet, then (9) turns out as follows:

$$\frac{\partial A_{z1}}{\mu_1 \partial r} + \mu_0 M_\theta = \frac{\partial A_{z2}}{\mu_2 \partial r}$$ (11)

In Figure 1b, the continuity of the radial components of the vector potential leads to twelve following boundary conditions.

$$r = R_7 \rightarrow \frac{\partial A_z^I}{\mu_r \partial r} + \mu_0 M_\theta = 0$$ (12)

$$r = R_6 \rightarrow \begin{cases} \frac{\partial A_z^I}{\mu_r \partial r} + \mu_0 M_\theta = \frac{\partial A_z^{II}}{\partial r} \\ A_z^{II} = A_z^I \end{cases}$$ (13)

$$r = R_5 \rightarrow \begin{cases} \frac{\partial A_z^{II}}{\partial r} = \sum_{p=1,2}^{P_m} \frac{\partial A_z^p}{\partial r} \\ A_z^p = A_z^{II} \end{cases}$$ (14)

$$r = R_4 \rightarrow \begin{cases} A_z^p = A_z^{III} \\ \frac{\partial A_z^{III}}{\partial r} = \sum_{p=1,2}^{P_m} \frac{\partial A_z^p}{\partial r} \end{cases}$$ (15)

$$r = R_3 \rightarrow \begin{cases} \frac{\partial A_z^{III}}{\partial r} = \sum_{i=1,2}^{Q} \frac{\partial A_z^i}{\partial r} \\ A_z^i = A_z^{III} \end{cases}$$ (16)

$$r = R_2 \rightarrow \begin{cases} A_z^i = A_z^j \\ \frac{\partial A_z^j}{\partial r} = \frac{\partial A_z^i}{\partial r} \end{cases}$$ (17)

$$r = R_1 \rightarrow \frac{\partial A_z^j}{\partial r} = 0$$ (18)

The coefficients were determined using a Fourier series expansion shown in Appendices A and B. The flux density at the air gap, then, is deduced from:

$$B_r = \frac{\partial A_z}{r \partial \theta}; \; B_\theta = -\frac{\partial A_z}{\partial r}$$ (19)

*2.3. The Maxwell Stress Tensor*

The Maxwell stress tensor is given by the following equation:

$$\sigma_{ij} = \frac{1}{\mu_0}\left(B_i B_j - \frac{1}{2}\delta_{ij}|\vec{B}|^2\right)$$

where

$$\delta_{ij} = \begin{bmatrix} 0 \leftrightarrow i \neq j \\ 1 \leftrightarrow i = j \end{bmatrix}$$ (20)

Applying 3D cylinder coordinates, Equation (20) turns out as such:

$$\sigma_{ij} = \begin{bmatrix} \sigma_{rr} & \sigma_{\theta r} & \sigma_{zr} \\ \sigma_{r\theta} & \sigma_{\theta\theta} & \sigma_{z\theta} \\ \sigma_{rz} & \sigma_{\theta z} & \sigma_{zz} \end{bmatrix} = \begin{bmatrix} \frac{B_r^2 - B_\theta^2}{2\mu_0} & \frac{B_\theta B_r}{\mu_0} & 0 \\ \frac{B_r B_\theta}{\mu_0} & \frac{-B_r^2 + B_\theta^2}{2\mu_0} & 0 \\ 0 & 0 & -\frac{B_r^2 + B_\theta^2}{2\mu_0} \end{bmatrix}$$ (21)

The force exerted over a surface S can be expressed as follows:

$$F = \int \sigma_{ij} dS_n = \int \begin{bmatrix} \frac{B_r^2 - B_\theta^2}{2\mu_0} & \frac{B_\theta B_r}{\mu_0} & 0 \\ \frac{B_r B_\theta}{\mu_0} & \frac{-B_r^2 + B_\theta^2}{2\mu_0} & 0 \\ 0 & 0 & -\frac{B_r^2 + B_\theta^2}{2\mu_0} \end{bmatrix} \begin{bmatrix} dS_r \\ dS_\theta \\ dS_z \end{bmatrix} = \begin{bmatrix} \int \frac{B_r^2 - B_\theta^2}{2\mu_0} dS_r + \int \frac{B_\theta B_r}{\mu_0} dS_\theta \\ \int \frac{B_r B_\theta}{\mu_0} dS_r + \int \frac{-B_r^2 + B_\theta^2}{2\mu_0} dS_\theta \\ \int -\frac{B_r^2 + B_\theta^2}{2\mu_0} dS_z \end{bmatrix} \quad (22)$$

Generally, the attraction force consists of three surface directions in the integral. However, along the $z$-axis, the force density is constant, resulting in a zero integral for the $z$-directional force. Additionally, when compared to the radial force density, the force density in the $\theta$-direction is relatively small and can be neglected. As a result, the average force density acting on the modulation can be simplified as follows:

$$\sigma = \begin{bmatrix} \sigma_r \\ \sigma_\theta \\ \sigma_z \end{bmatrix} = \begin{bmatrix} \left.\frac{B_r^2 - B_\theta^2}{2\mu_0}\right|_{R_5} - \frac{R_4}{R_5}\left.\frac{B_r^2 - B_\theta^2}{2\mu_0}\right|_{R_4} \\ \left.\frac{B_r B_\theta}{\mu_0}\right|_{R_5} - \frac{R_4}{R_5}\left.\frac{B_r B_\theta}{\mu_0}\right|_{R_4} \\ 0 \end{bmatrix} \quad (23)$$

As shown in Figure 8b, during the mechanical modulation analysis, the tangential component of the average magnetic force density appears notably smaller compared to the radial counterpart, often rendering it negligible in the analysis.

The electromagnetic torque is derived using the Maxwell stress tensor. An integration path along a circle with a radius within the air gap subdomain is employed, leading to the electromagnetic torque presentation as detailed in [35].

## 3. Stress and Deformation Analysis in Modulation

### 3.1. Timoshenko Beam Model

In the beginning, in order to simplify the model shown in Figure 5, the following assumptions are made:

- The coordinate $Or\theta z$ is converted into $Oxyz$;
- The cross section and material properties of the beam are constant along the length;
- There is a symmetrical cross-section about x-y plane;
- The height, width, and length of a converted bar are denoted as X, Y, and Z.

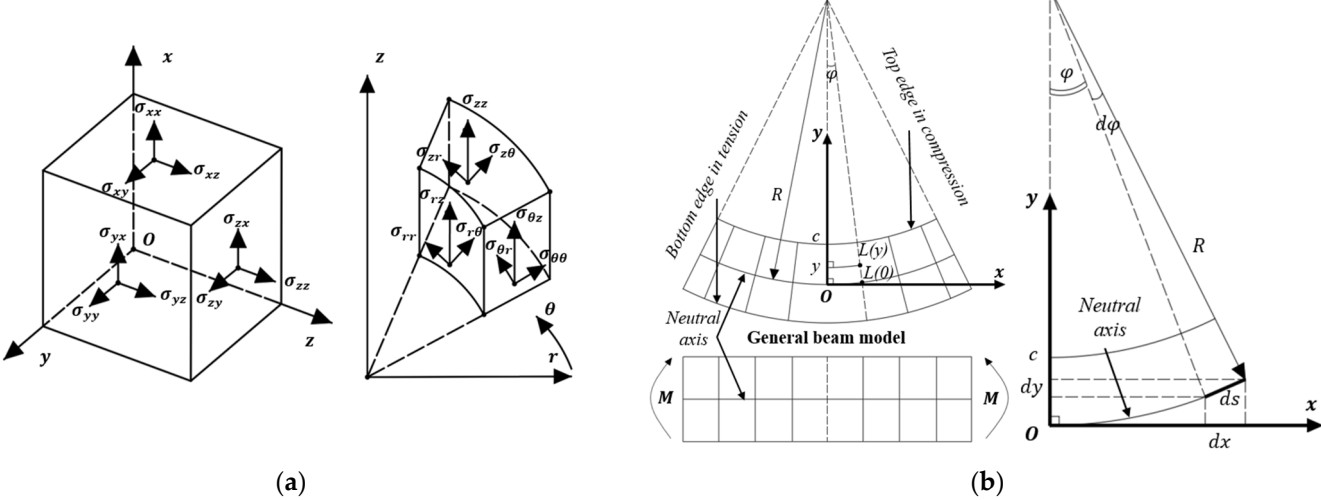

**Figure 5.** (**a**) The stress tensor in a 3D space $Oxyz$ and $Or\theta z$. (**b**) The general beam.

The model depicted in Figure 5b is commonly known as the Euler–Bernoulli beam, which assumes that both sides of the beam are free. However, in reality, in machines,

one or both sides of the beam shown in Figure 6 may be fixed. This difference between mathematical computation and finite element method (FEM) simulation introduces a significant error, especially when the $y$-axis thickness ($Y$) is sufficiently large.

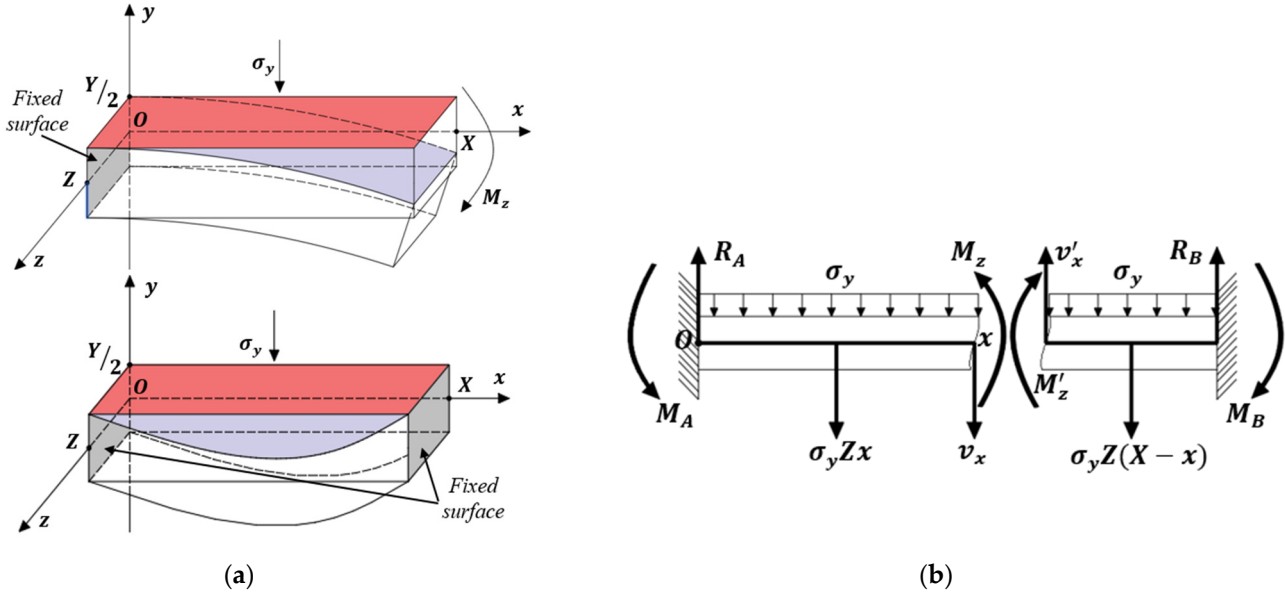

**Figure 6.** (**a**) An illustration of the modulation being deformed one side and two sides and (**b**) their simplified model for the mathematical solution.

To address this problem, a Timoshenko beam model is introduced in [42]. The Timoshenko beam model takes into account the effect of shear deformation, providing a more accurate representation of the actual behavior of beams with fixed ends. Through incorporating the Timoshenko beam model, the error caused by the assumption of free ends in the Euler–Bernoulli beam model can be minimized. When applying a moment, $M_z$, on a surface area, $A$, and coefficients are given as in Table 1, the PDEs of the beam model are expressed as follows:

$$\begin{cases} \frac{dy}{dx} = \theta_x + \frac{1}{\kappa AG} \frac{dM_z}{dx} \\ \frac{d\theta_x}{dx} = -\frac{M_z}{EI_z} \end{cases} \tag{24}$$

where $\theta_x$ is the angle of rotation of the normal to the mid-surface of the beam, and the second moment of area, $I_z$, is calculated similarly for both cases in Figure 6.

$$I_z = \frac{Y^3 Z}{12} \tag{25}$$

*3.2. Modulation Fixed at One End*

Considering an arbitrary point along the beam that divides it into two sections, as shown in Figure 6b, our focus lies on the case where one side of the beam is fixed. In this scenario, we exclusively examine the left-side portion. The distribution of the moment acting along the modulation can be described using the following equation:

$$M_{zx} = \sigma_y Z \frac{-(X-x)^2}{2} \tag{26}$$

Substituting (26) into (24), then combining the first boundary condition $\theta_{x=0} = 0$, the angle, $\theta_x$, is derived as:

$$\theta_x = -\frac{\sigma_y Z}{EI_z} \frac{(X-x)^3}{6} + \frac{\sigma_y Z}{EI_z} \frac{X^3}{6} \tag{27}$$

Through applying the second boundary condition $y_{x=0} = 0$ for Equation (24), the deformation along the length of the modulation can be defined as follows:

$$y = \frac{\sigma_y Z}{EI_z}\frac{(X-x)^4}{24} + \frac{\sigma_y Z}{EI_z}\frac{X^3 x}{6} + \frac{\sigma_y Z}{\kappa AG}\frac{-(X-x)^2}{2} - \frac{\sigma_y Z}{EI_z}\frac{X^4}{24} + \frac{\sigma_y Z}{\kappa AG}\frac{X^2}{2} \tag{28}$$

Under uniform stress acting in the $y$-direction, the stress distribution in the $Oxy$ plane is calculated as:

$$\sigma_y = \frac{yM_{zx}}{I_z} = -y\sigma_y Z\frac{(X-x)^2}{2} \tag{29}$$

*3.3. Modulation Fixed at Both Ends*

In the scenario of modulation fixed at both ends, both sections are required. Similar to Section 3.2, the following expressions apply:

$$M_{zx} = \sigma_y Z\left(\frac{Xx}{2} - \frac{x^2}{2} - \frac{X^2}{12}\right) \tag{30}$$

$$\theta_x = \frac{\sigma_y Z}{EI_z}\left(\frac{x^3}{6} - \frac{Xx^2}{4} + \frac{X^2 x}{12}\right) \tag{31}$$

Notably, because of symmetry along the center of the beam, the conditions, therefore, become $\theta_{x=\{0;X\}} = 0$ and $y_{x=\{0;X\}} = 0$. The deformation and stress of modulation are given as follows:

$$y = \frac{\sigma_y Z}{EI_z}\frac{(X-x)^2 x^2}{24} + \frac{\sigma_y Z}{\kappa AG}\frac{x(X-x)}{2} \tag{32}$$

$$\sigma_y = \frac{yM_{zx}}{I_z} = \frac{y\sigma_y Z}{I_z}\left(\frac{Xx}{2} - \frac{x^2}{2} - \frac{X^2}{12}\right) \tag{33}$$

## 4. Results and Discussion

An example, with parameters detailed in Table 2 and material characteristics presented in Table 1, was subjected to FEM simulation and analytical analysis. Notably, the FEM mesh for both electromagnetic and mechanical analyses is illustrated in Figure 7a.

The electromagnetic torque, air gap flux densities, and magnetic force density obtained through the analytical method aligned closely with FEM simulation results, as depicted in Figures 7b–d and 8a. The tangential component of the average magnetic force density is approximately zero in Figure 8b. This leads to the neglect of tangential components in stress analysis in this study. The non-uniform force distribution is apparent in these figures, contributing to uneven deformation in the modulations.

Figure 9 shows non-uniform deformation as well as stress in bars. The first bar has almost no significant influence from the magnetic force, whereas an extreme bend is occupied in the ninth bar.

Generally, there is a strong agreement between the mathematical method and FEM results, as depicted in Figure 10, which presents deformation and stress profiles. In the case of two fixed ends (depicted in Figure 10b), although the stress distribution along the bar is not entirely consistent, the maximum stress at the four corners is accurately predicted. Notably, the scenario with two fixed ends exhibits greater stability in comparison to the case with only one fixed end. In the latter, the maximum deformation and stress are approximately 1/20 and 1/8 times larger, respectively.

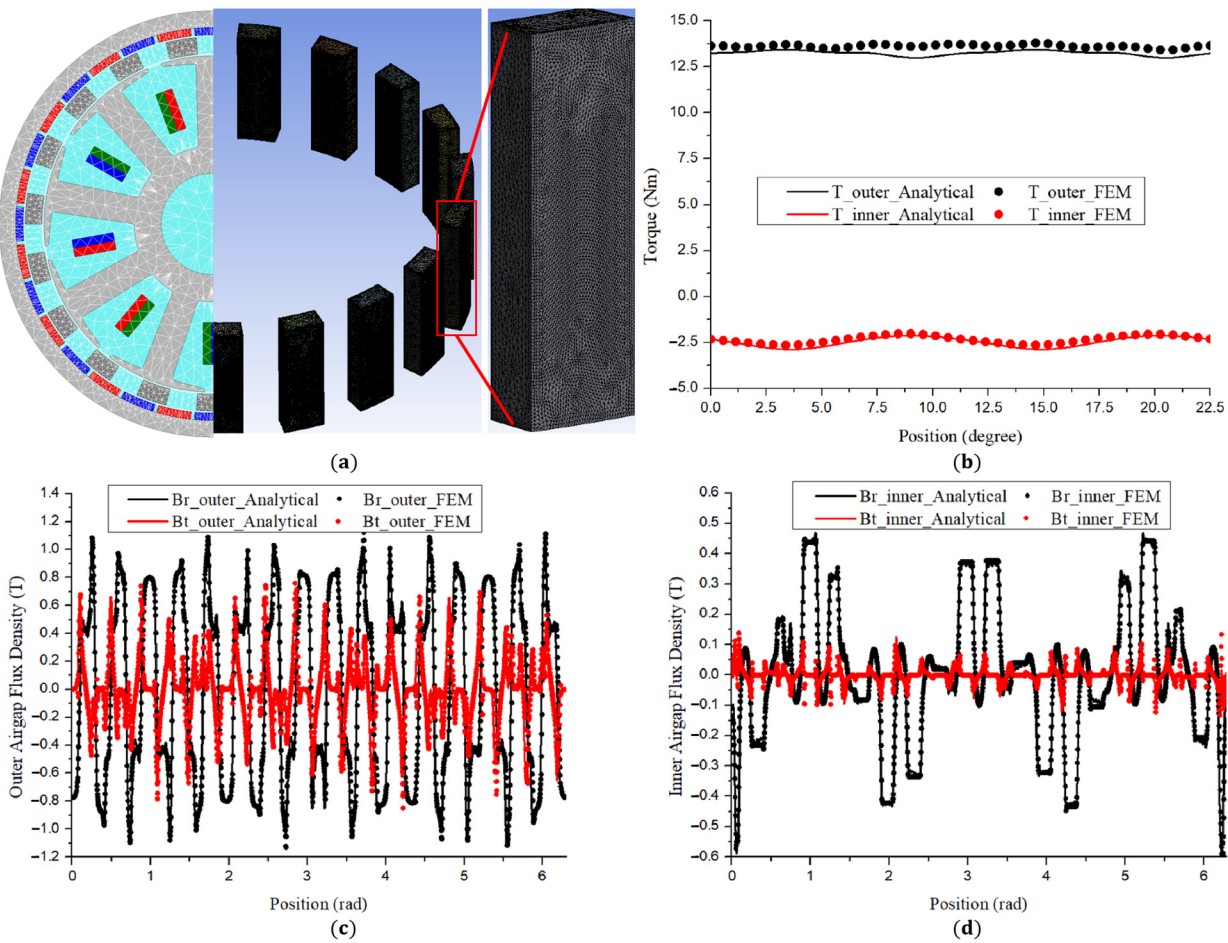

**Figure 7.** (**a**) A 2D model for magnetic attraction force analysis (33,450 mesh elements) and a 3D model for stress/deformation analysis (632,268 mesh elements); (**b**) the electromagnetic torque in the outer air gap and inner air gap; the air gap flux density in the (**c**) outer air gap and (**d**) inner air gap.

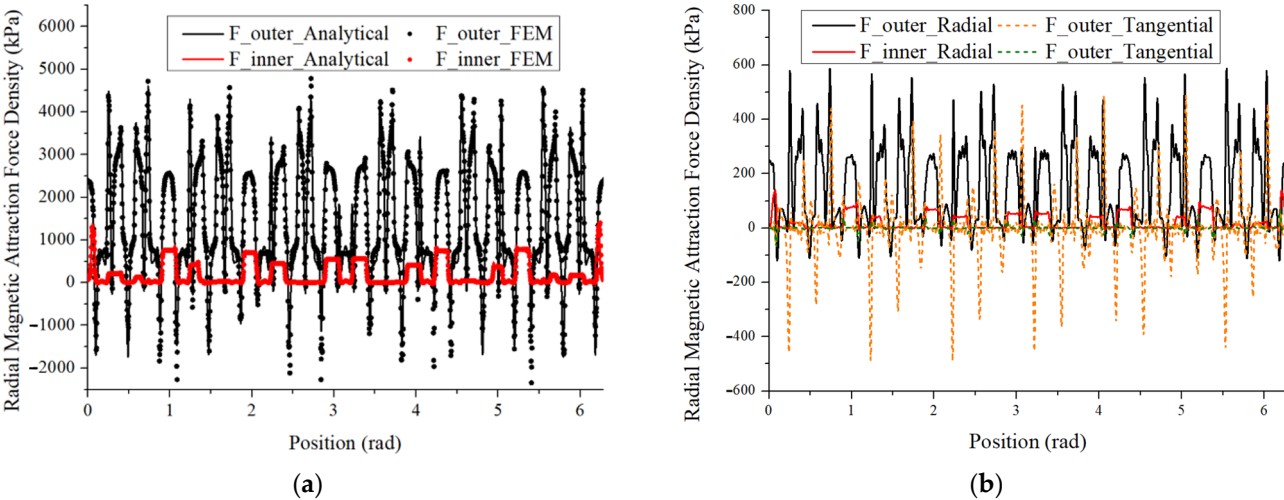

**Figure 8.** (**a**) The radial magnetic attraction force density; (**b**) the radial and tangential magnetic force density in the outer and inner air gap.

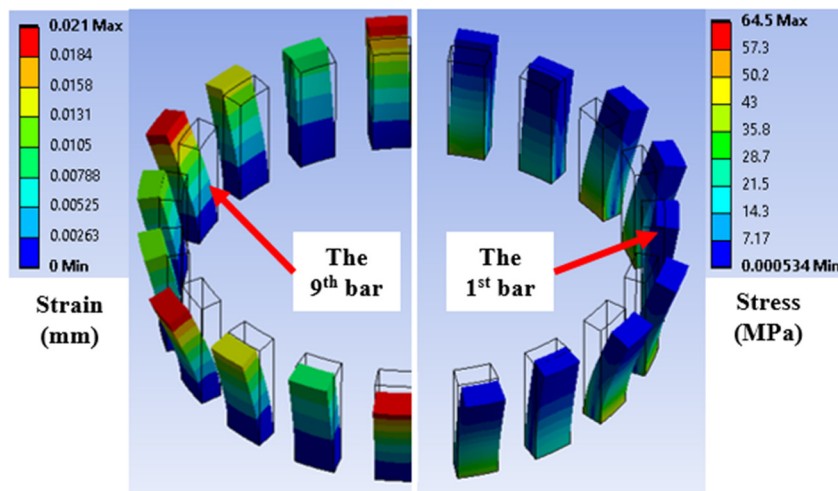

**Figure 9.** Mechanical analysis using the FEM of the model with one end fixed.

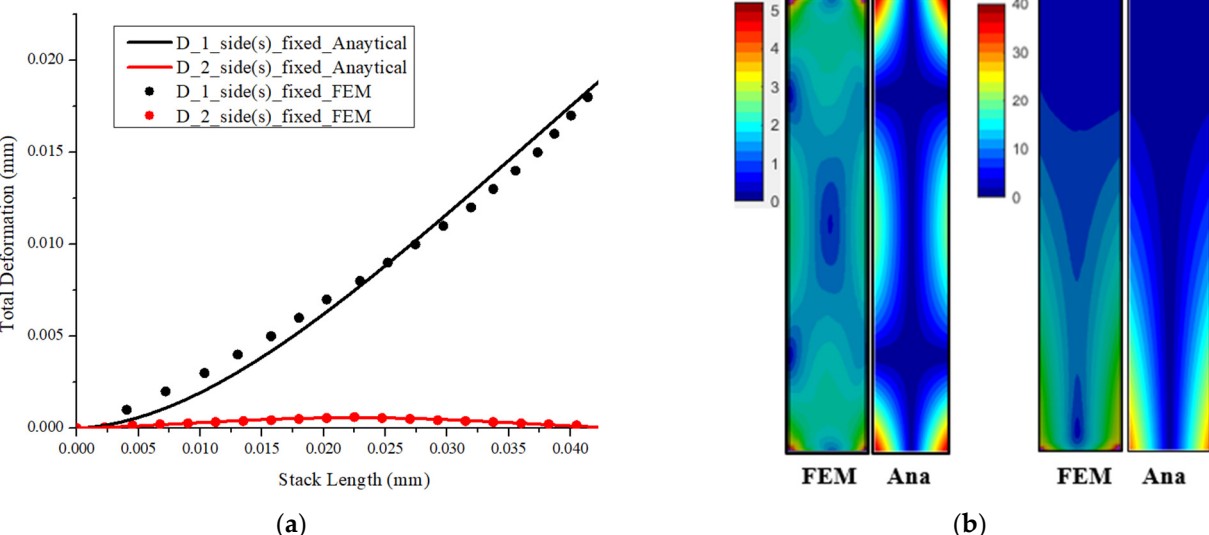

**Figure 10.** The ninth modulation bar comparison between analytical method and FEM simulation of (**a**) deformation, stress of model with (**b**) one end fixed (right), and both ends fixed (left).

## 5. Conclusions

The proposed mathematical method was an efficient approach for obtaining mechanical characteristics in the modulation. Moreover, compared with 2D magnetic attraction force and 3D stress/deformation FEM simulations, the computational time of the mathematical method was reduced from 30 s and 5 min to 2 s and 1.5 s, respectively. This allows designers to apply optimization techniques that save time exponentially.

Continuing research based on this paper can be outlined as follows:

- Consider the non-linear characteristic of magnetic material;
- Consider the skin effect for the solid modulation;
- The influence of the on-load or unbalanced load should be ideal;
- Verify the subdomain method with the test bench's results;
- Develop a mathematical method for a lamination-structured modulation.

**Author Contributions:** Conceptualization, M.-D.N.; methodology, M.-D.N., K.-H.S., Y.-J.K., A.-T.P. and J.-Y.C.; software, M.-D.N.; validation, M.-D.N.; formal analysis, M.-D.N.; investigation, M.-D.N.; resources, J.-Y.C.; data curation, M.-D.N.; writing—original draft preparation, M.-D.N.; writing —review and editing, S.-M.K., J.-I.L., H.-S.S., Y.-K.L., H.-K.L., K.-H.S., Y.-J.K., A.-T.P. and J.-Y.C.; visualization, M.-D.N. and S.-M.K.; supervision, K.-H.S., Y.-J.K., A.-T.P. and J.-Y.C.; project administration, J.-Y.C.; funding acquisition, J.-Y.C. All authors have read and agreed to the published version of the manuscript.

**Funding:** This work was supported by Korea Institute of Planning and Evaluation for Technology in Food, Agriculture and Forestry(IPET) through Technology Commercialization Support Program, funded by Ministry of Agriculture, Food and Rural Affairs(MAFRA)(821014-03 and 122047-03).

**Data Availability Statement:** Not applicable.

**Conflicts of Interest:** The authors declare no conflict of interest.

## Appendix A

The following integrals are presented to shorten equations in Appendix B:

$$sni(n, \theta_i, \beta) = \int_{\theta_i}^{\theta_i+\beta} \sin(n\theta)d\theta = \frac{1}{n}(\cos(n\theta_i) - \cos(n(\theta_i + \beta))) \tag{A1}$$

$$rni(n, \theta_i, \beta) = \int_{\theta_i}^{\theta_i+\beta} \cos(n\theta)d\theta = \frac{1}{n}(\sin(n(\theta_i + \beta)) - \sin(n\theta_i)) \tag{A2}$$

$$gkni(k, n, \theta_i, \beta) = \int_{\theta_i}^{\theta_i+\beta} \sin(n\theta)\cos\left(k\frac{\pi}{\beta}(\theta - \theta_i)\right)d\theta$$
$$= \begin{cases} \frac{\beta}{2}\left(\sin(n\theta_i) - \frac{1}{2k\pi}(\cos(n(\theta_i + 2\beta)) - \cos(n\theta_i))\right) \leftrightarrow k\pi = n\beta \\ \frac{n\beta^2}{(k\pi)^2 - (n\beta)^2}\left((-1)^k(n(\theta_i + 2\beta)) - \cos(n\theta_i)\right) \leftrightarrow k\pi \neq n\beta \end{cases} \tag{A3}$$

$$fkni(k, n, \theta_i, \beta) = \int_{\theta_i}^{\theta_i+\beta} \cos(n\theta)\cos\left(k\frac{\pi}{\beta}(\theta - \theta_i)\right)d\theta$$
$$= \begin{cases} \frac{\beta}{2}\left(\cos(n\theta_i) + \frac{1}{2k\pi}(\sin(n(\theta_i + 2\beta)) - \sin(n\theta_i))\right) \leftrightarrow k\pi = n\beta \\ \frac{-n\beta^2}{(k\pi)^2 - (n\beta)^2}\left((-1)^k\sin(n(\theta_i + 2\beta)) - \sin(n\theta_i)\right) \leftrightarrow k\pi \neq n\beta \end{cases} \tag{A4}$$

$$Fmk(m, k, \beta, \delta) = \int_{\theta_i}^{\theta_i+\beta} \cos\left(k\frac{\pi}{\beta}(\theta - \theta_i)\right)\cos\left(m\frac{\pi}{\delta}(\theta - \theta_j)\right)d\theta$$
$$= \begin{cases} \frac{\beta}{2}\cos\left(\frac{k\pi}{2\beta}(\beta - \delta)\right) \leftrightarrow \frac{m}{\delta} = \frac{k}{\beta} \\ \frac{m\frac{\pi}{\delta}}{\left(m\frac{\pi}{\delta}\right)^2 - \left(k\frac{\pi}{\beta}\right)^2}\left((-1)^k\sin\left(\frac{m\pi}{2\delta}(\beta + \delta)\right) + \sin\left(\frac{m\pi}{2\delta}(\beta - \delta)\right)\right) \leftrightarrow \frac{m}{\delta} \neq \frac{k}{\beta} \end{cases} \tag{A5}$$

## Appendix B

From (1) to (8), vector potential in sub-domains can be rewritten as:

$$A_z^I = \sum_{n=1,2}^{\infty} \left(r^n A_n^I + r^{-n}B_n^I + P_n\sin(n\theta_0)\right)\sin(n\theta) + \left(r^n C_n^I + r^{-n}D_n^I + P_n\cos(n\theta_0)\right)\cos(n\theta) \tag{A6}$$

$$A_z^{II} = \sum_{n=1,2}^{\infty} \left(r^n A_n^{II} + r^{-n}B_n^{II}\right)\sin(n\theta) + \left(r^n C_n^{II} + r^{-n}D_n^{II}\right)\cos(n\theta) \tag{A7}$$

$$A_z^p = A_0^p + ln(r)B_0^p + \sum_{h=1,2}^{\infty} \left( r^{h\frac{\pi}{\alpha}} C_h^p + r^{-h\frac{\pi}{\alpha}} D_h^p \right) \cos\left( h\frac{\pi}{\alpha}(\theta - \theta_p) \right) \tag{A8}$$

$$A_z^{III} = \sum_{n=1,2}^{\infty} \left( r^n A_n^{III} + r^{-n} B_n^{III} \right) \sin(n\theta) + \left( r^n C_n^{III} + r^{-n} D_n^{III} \right) \cos(n\theta) \tag{A9}$$

$$A_z^i = A_0^i + ln(r)B_0^i + \sum_{k=1,2}^{\infty} \left( r^{k\frac{\pi}{\beta}} C_k^i + r^{-k\frac{\pi}{\beta}} D_k^i \right) \cos\left( k\frac{\pi}{\beta}(\theta - \theta_i) \right) \tag{A10}$$

$$A_z^j = A_0^j + ln(r)B_0^j - \frac{\mu_0}{4}r^2 J_0^j + \sum_{m=1,2}^{\infty} \left( r^{m\frac{\pi}{\delta}} C_m^j + r^{-m\frac{\pi}{\delta}} D_m^j + \frac{\mu_0}{\left(m\frac{\pi}{\delta}\right)^2 - 4}r^2 J_m^j \right) \cos\left( m\frac{\pi}{\delta}(\theta - \theta_j) \right) \tag{A11}$$

where

$$\left[ \begin{array}{l} P_n = \frac{\mu_0 r}{n^2-1}(-M_{\theta n}^I + nM_{rn}^I) \leftrightarrow n \neq 1 \\ P_n = 0.5\mu_0 \ln(r)r(-M_{\theta n}^I + nM_{rn}^I) \leftrightarrow n = 1 \end{array} \right.$$

Based on the above forms of vector potential and boundary conditions presented from Equations (12) to (18), coefficients are calculated via solving the following equation system:

$$\frac{1}{\mu_r} \left( R_7^{n-1} nA_n^I - nR_7^{-n-1} B_n^I + P_n' \sin(n\theta_0) \right) + \mu_0 M_{\theta n}^I \sin(n\theta_0) = 0 \tag{A12}$$

$$\frac{1}{\mu_r} \left( R_7^{n-1} nC_n^I - nR_7^{-n-1} D_n^I + P_n' \cos(n\theta_0) \right) + \mu_0 M_{\theta n}^I \cos(n\theta_0) = 0 \tag{A13}$$

$$\frac{1}{\mu_r} \left( R_6^{n-1} nA_n^I - nR_6^{-n-1} B_n^I + P_n' \sin(n\theta_0) \right) + \mu_0 M_{\theta n}^I \sin(n\theta_0) = \left( R_6^{n-1} nA_n^{II} - nR_6^{-n-1} B_n^{II} \right) \tag{A14}$$

$$\frac{1}{\mu_r} \left( R_6^{n-1} nC_n^I - nR_6^{-n-1} D_n^I + P_n' \cos(n\theta_0) \right) + \mu_0 M_{\theta n}^I \cos(n\theta_0) = \left( R_6^{n-1} nC_n^{II} - nR_6^{-n-1} D_n^{II} \right) \tag{A15}$$

$$R_6^n A_n^{II} + R_6^{-n} B_n^{II} = R_6^n A_n^I + R_6^{-n} B_n^I + P_n \sin(n\theta_0) \tag{A16}$$

$$R_6^n C_n^{II} + R_6^{-n} D_n^{II} = R_6^n C_n^I + R_6^{-n} D_n^I + P_n \cos(n\theta_0) \tag{A17}$$

$$R_5^{n-1} nA_n^{II} - nR_5^{-n-1} B_n^{II} = \sum_{p=1,2}^{P_m} \left( \frac{B_0^p}{R_5} \frac{1}{\pi} sni(n, \theta_p, \gamma) + \sum_{h=1,2}^{\infty} \left( h\frac{\pi}{\alpha} \left( R_5^{h\frac{\pi}{\alpha}-1} C_h^p - R_5^{-h\frac{\pi}{\alpha}-1} D_h^p \right) \frac{1}{\pi} gkni(k, n, \theta_p, \gamma) \right) \right) \tag{A18}$$

$$R_5^{n-1} nC_n^{II} - nR_5^{-n-1} D_n^{II} = \sum_{p=1,2}^{P_m} \left( \frac{B_0^p}{R_5} \frac{1}{\pi} rni(n, \theta_p, \gamma) + \sum_{h=1,2}^{\infty} \left( h\frac{\pi}{\alpha} \left( R_5^{h\frac{\pi}{\alpha}-1} C_h^p - R_5^{-h\frac{\pi}{\alpha}-1} D_h^p \right) \frac{1}{\pi} fkni(k, n, \theta_p, \gamma) \right) \right) \tag{A19}$$

$$R_5^{h\frac{\pi}{\alpha}} C_h^p + R_5^{-h\frac{\pi}{\alpha}} D_h^p = \sum_{n=1,2}^{\infty} \left( R_5^n A_n^{II} - R_5^{-n} B_n^{II} \right) \frac{2}{\gamma} gkni(k, n, \theta_p, \gamma) + \left( R_5^n C_n^{II} - R_5^{-n} D_n^{II} \right) \frac{2}{\gamma} fkni(k, n, \theta_p, \gamma) \tag{A20}$$

$$A_0^p + ln(R_5)B_0^p = \sum_{n=1,2}^{\infty} \left( R_5^n A_n^{II} - R_5^{-n} B_n^{II} \right) \frac{1}{\gamma} sni(n, \theta_p, \gamma) + \left( R_5^n C_n^{II} - R_5^{-n} D_n^{II} \right) \frac{1}{\gamma} rni(n, \theta_p, \gamma) \tag{A21}$$

$$R_4^n A_n^{III} + R_4^{-n} B_n^{III} = \sum_{p=1,2}^{P_m} \left( \frac{B_0^p}{R_4} \frac{1}{\pi} sni(n, \theta_p, \gamma) + \sum_{h=1,2}^{\infty} h\frac{\pi}{\alpha} \left( R_4^{h\frac{\pi}{\alpha}-1} C_h^p - R_4^{-h\frac{\pi}{\alpha}-1} D_h^p \right) \frac{1}{\pi} gkni(k, n, \theta_p, \gamma) \right) \tag{A22}$$

$$R_4^n C_n^{III} + R_4^{-n} D_n^{III} = \sum_{p=1,2}^{P_m} \left( \frac{B_0^p}{R_4} \frac{1}{\pi} rni(n,\theta_p,\gamma) + \sum_{h=1,2}^{\infty} h\frac{\pi}{\alpha} \left( R_4^{h\frac{\pi}{\alpha}-1} C_h^p - R_4^{-h\frac{\pi}{\alpha}-1} D_h^p \right) \frac{1}{\pi} fkni(k,n,\theta_p,\gamma) \right) \tag{A23}$$

$$R_4^{h\frac{\pi}{\alpha}} C_h^p + R_4^{-h\frac{\pi}{\alpha}} D_h^p = \sum_{n=1,2}^{\infty} \left( R_4^n A_n^{III} - R_4^{-n} B_n^{III} \right) \frac{2}{\gamma} gkni(k,n,\theta_p,\gamma) + \left( R_4^n C_n^{III} - R_4^{-n} D_n^{III} \right) \frac{2}{\gamma} fkni(k,n,\theta_p,\gamma) \tag{A24}$$

$$A_0^p + ln(R_4) B_0^p = \sum_{n=1,2}^{\infty} \left( R_4^n A_n^{III} - R_4^{-n} B_n^{III} \right) \frac{1}{\gamma} sni(n,\theta_p,\gamma) + \left( R_4^n C_n^{III} - R_4^{-n} D_n^{III} \right) \frac{1}{\gamma} rni(n,\theta_p,\gamma) \tag{A25}$$

$$R_3^{n-1} n A_n^{III} - R_3^{-n-1} n B_n^{III} = \sum_{i=1,2}^{Q} \left( \frac{B_0^i}{R_3} \frac{1}{\pi} sni(n,\theta_i,\beta) + \sum_{k=1,2}^{\infty} \left( k\frac{\pi}{\beta} \left( R_3^{k\frac{\pi}{\beta}-1} C_k^i - R_3^{-k\frac{\pi}{\beta}-1} D_k^i \right) \frac{1}{\pi} gkni(k,n,\theta_i,\beta) \right) \right) \tag{A26}$$

$$R_3^{n-1} n C_n^{III} - R_3^{-n-1} n D_n^{III} = \sum_{i=1,2}^{Q} \left( \frac{B_0^i}{R_3} \frac{1}{\pi} rni(n,\theta_i,\beta) + \sum_{k=1,2}^{\infty} \left( k\frac{\pi}{\beta} \left( R_3^{k\frac{\pi}{\beta}-1} C_k^i - R_3^{-k\frac{\pi}{\beta}-1} D_k^i \right) \frac{1}{\pi} fkni(k,n,\theta_i,\beta) \right) \right) \tag{A27}$$

$$R_3^{k\frac{\pi}{\beta}} C_k^i + R_3^{-k\frac{\pi}{\beta}} D_k^i = \sum_{n=1,2}^{\infty} \left( R_3^n A_n^{III} - R_3^{-n} B_n^{III} \right) \frac{2}{\beta} gkni(k,n,\theta_i,\beta) + \left( R_3^n C_n^{III} - R_3^{-n} D_n^{III} \right) \frac{2}{\beta} fkni(k,n,\theta_i,\beta) \tag{A28}$$

$$A_0^i + ln(R_3) B_0^i = \sum_{n=1,2}^{\infty} \left( R_3^n A_n^{III} - R_3^{-n} B_n^{III} \right) \frac{1}{\beta} sni(n,\theta_i,\beta) + \left( R_3^n C_n^{III} - R_3^{-n} D_n^{III} \right) \frac{1}{\beta} rni(n,\theta_i,\beta) \tag{A29}$$

$$R_2^{k\frac{\pi}{\beta}} C_k^i + R_2^{-k\frac{\pi}{\beta}} D_k^i = \sum_{m=1,2}^{\infty} \left( R_2^{m\frac{\pi}{\delta}} C_m^j + R_2^{-m\frac{\pi}{\delta}} D_m^j + \frac{\mu_0}{\left(m\frac{\pi}{\delta}\right)^2 - 4} R_3^2 J_m^j \right) \frac{2}{\beta} Fmk(m,k,\beta,\delta) \tag{A30}$$

$$
\begin{aligned}
A_0^i + ln(R_2) B_0^i = &\, A_0^j + ln(R_2) B_0^j - \frac{\mu_0}{4} R_2^2 J_0^j \\
&+ \frac{\delta}{\beta m\pi} \sum_{m=1,2}^{\infty} \left( R_2^{m\frac{\pi}{\delta}} C_m^j + R_2^{-m\frac{\pi}{\delta}} D_m^j + \frac{\mu_0}{\left(m\frac{\pi}{\delta}\right)^2-4} R_2^2 J_m^j \right) \left( \sin\left(\frac{m\pi\beta}{\delta}\right) - \sin\left(\frac{m\pi}{\delta}(\theta_i - \theta_j)\right) \right)
\end{aligned}
\tag{A31}$$

$$
\begin{aligned}
m\frac{\pi}{\delta} &\left( R_2^{m\frac{\pi}{\delta}-1} C_m^j - R_2^{-m\frac{\pi}{\delta}-1} D_m^j \right) + \frac{2 R_2 J_m^j \mu_0}{\left(m\frac{\pi}{\delta}\right)^2 - 4} \\
&= \frac{2}{\beta m\pi} \frac{1}{R_2} B_0^i \left( \sin\left(\frac{m\pi\beta}{\delta}\right) - \sin\left(\frac{m\pi}{\delta}(\theta_i - \theta_j)\right) \right) + \sum_{k=1,2}^{\infty} k\frac{\pi}{\beta} \left( R_2^{k\frac{\pi}{\beta}-1} C_k^i - R_2^{-k\frac{\pi}{\beta}-1} D_k^i \right) \frac{2}{\delta} Fmk(m,k,\beta,\delta)
\end{aligned}
\tag{A32}$$

$$\frac{1}{R_2} B_0^j - \frac{\mu_0}{2} R_2 J_0^j = \frac{\beta}{\delta} \frac{1}{R_2} B_0^i \tag{A33}$$

$$R_1^{m\frac{\pi}{\delta}-1} m\frac{\pi}{\delta} C_m^j - R_1^{-m\frac{\pi}{\delta}-1} m\frac{\pi}{\delta} D_m^j + \frac{2 R_2 J_m^j \mu_0}{\left(m\frac{\pi}{\delta}\right)^2 - 4} = 0 \tag{A34}$$

$$\frac{1}{R_1} B_0^j - \frac{\mu_0}{2} R_1 J_0^j = 0 \tag{A35}$$

Through reformatting the given equations into a matrix and vector, we can employ mathematical software (MATLAB) to find a numerical solution.

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
