# Peer review of "Prediction of Stress and Deformation Caused by Magnetic Attraction Force in Modulation Elements in a Magnetically Geared Machine Using Subdomain Modeling"

_machines, doi:10.3390/machines11090887_

Round 1
Reviewer 1 Report
The article proposes a simplified analytical method for analyzing the deformation of magnetic circuit elements caused by the forces of attraction of a magnetic nature in a magnetically geared Machine. The magnetic field is assumed to be two-dimensional. The non-linearity of the magnetic core, the movement and the influence of strain on the change in the magnetic field distribution were omitted. There is no information in the assumptions about not taking into account eddy currents in conductive ferromagnetic elements. The consistency of the results of analytical calculations with the results obtained using FEM was confirmed. In my opinion, the article is formally written correctly.
The article lacks a broader discussion on the purposefulness of using the proposed approach in practice. Only analytical solutions with FEM were compared. The indicated advantage is only the shortened calculation time in relation to 2D FEM calculations. There is no debate about the accuracy of 2D magnetic field determination with 3D distribution. Is the 2D approach sufficient? Why is there no information about deformations of magnetic circuit elements in the circumferential direction. Are they negligible? The static state is analyzed - there is no movement. There is no information about the influence of ferromagnetic nonlinearity and eddy currents accompanying the motion on the actual distribution of the magnetic field and stress. For the above reasons, the conclusions are poor. The conclusions mention that further work is planned on the analysis of the impact of the load on the operation of the system.
Detailed notes. No explanation of the markings used: Zp in equation (7) and in (Fig. 3); parameters α, Mo - equation (7); parameters E, κ, G - equation (24), parameters Y, Z - equation (25) and X - equation (26). Some of the above designations are explained only in Chapter 4 in Table 2.
Reviewer 2 Report
The research content of the article has certain significance, but it needs to be greatly improved. The specific modifications are as follows:
1. Regarding the analytical method, this article lacks important references, such as the articles of Professor Zhu Ziqiang and Terry Lubin, and the author must add their articles as references.
2. Regarding the explanation of magnetic gears and magnetic gear motors in the introduction section, the author should have the latest references as support, such as Design and Optimization of Coaxial Magnetic Gear with Double Layer PMs and Spoke Structure for Tidal Power Generation, Characteristic Analysis of the Magnetic Variable Speed Diesel Electric Hybrid Motor with Auxiliary Teeth for Ship Propulsion
3. What about torque calculation? The author should provide the calculation results;
4. How about calculating the time? There should be comparative analysis;
5. What is the dynamic performance of the motor? Provide explanations or provide experimental results.
Need to improve it.
Round 2
Reviewer 2 Report
Thank you for your reply. Please modify the paper format, especially the reference format, according to the requirements of the journal!
No.Author Response
Dear reviewer,
Thank you so much for your response,
I did modify my paper format, especially the reference's appearance, bulleted list, and reference format.
Thank you so much for your consideration.
